# Quantitative and multiplexed chemical-genetic phenotyping in mammalian cells with QMAP-Seq

Sonia Brockway [1,2,3,4], Geng Wang [1,2,3], Jasen M. Jackson[1,2,3], David R. Amici [1,2,3,5],
Seesha R. Takagishi[1,2,3], Matthew R. Clutter[3,6,7], Elizabeth T. Bartom [1,2] & Marc L. Mendillo [1,2,3✉]

Chemical-genetic interaction profiling in model organisms has proven powerful in providing insights into compound mechanism of action and gene function. However, identifying chemical-genetic interactions in mammalian systems has been limited to low-throughput or computational methods. Here, we develop Quantitative and Multiplexed Analysis of Phenotype by Sequencing (QMAP-Seq), which leverages next-generation sequencing for pooled high-throughput chemical-genetic profiling. We apply QMAP-Seq to investigate how cellular stress response factors affect therapeutic response in cancer. Using minimal automation, we treat pools of 60 cell types—comprising 12 genetic perturbations in five cell lines—with 1440 compound-dose combinations, generating 86,400 chemical-genetic measurements. QMAP-Seq produces precise and accurate quantitative measures of acute drug response comparable to gold standard assays, but with increased throughput at lower cost. Moreover, QMAP-Seq reveals clinically actionable drug vulnerabilities and functional relationships involving these stress response factors, many of which are activated in cancer. Thus, QMAP-Seq provides a broadly accessible and scalable strategy for chemical-genetic profiling in mammalian cells.

[1] Department of Biochemistry and Molecular Genetics, Northwestern University Feinberg School of Medicine, Chicago, IL 60611, USA. [2] Simpson Querrey Institute for Epigenetics, Northwestern University Feinberg School of Medicine, Chicago, IL 60611, USA. [3] Robert H. Lurie Comprehensive Cancer Center, Northwestern University Feinberg School of Medicine, Chicago, IL 60611, USA. [4] Driskill Graduate Program in Life Sciences, Northwestern University Feinberg School of Medicine, Chicago, IL 60611, USA. [5] Medical Scientist Training Program, Northwestern University Feinberg School of Medicine, Chicago, IL 60611, USA. [6] Chemistry of Life Processes Institute, Northwestern University, Evanston, IL 60208, USA. [7] Department of Molecular Biosciences, Northwestern University, Evanston, IL 60208, USA. ✉email: mendillo@northwestern.edu

Chemical–genetic interaction profiling in model organisms, such as yeast, has emerged as a powerful strategy to reveal functional insights into compounds, genes, and cellular processes. In these studies, the mechanism of action of a compound can be deduced by comparing its chemical–genetic interaction profile (the quantitative landscape of the effects of a panel of individual genes on the efficacy of this particular compound) to the profiles of compounds with known cellular targets to identify the most similar profiles[1–5]. Likewise, the function of a gene can be inferred by comparing its chemical-genetic interaction profile (the quantitative landscape of the effects of this particular gene on the efficacy of a panel of compounds) to the profiles of genes with known functions[6].

The development of highly specific and efficient genetic perturbation tools based on CRISPR-Cas9 has enabled similar types of chemical–genetic studies in mammalian systems, albeit at much smaller scales than in model organisms. Most often, chemical–genetic studies in mammalian systems involve genome-scale loss-of-function screens against one compound over the course of several weeks to identify drug targets[7] and define mechanisms of drug resistance[8,9]. Even focused studies interrogating limited numbers of chemical–genetic interactions can reveal critical insights. For example, one recent study demonstrated that the efficacy of a handful of clinical compounds was unaffected by knockout of their putative targets[10], highlighting the power of using chemical–genetic approaches to validate on-target activity of drug candidates. Despite their utility, these studies are low-throughput and thus limited to investigating small numbers of compounds.

There is a growing interest in identifying synthetic lethal and synthetic rescue chemical–genetic interactions that can serve as the basis for cancer therapeutic strategies. Chemical–genetic synthetic lethality, a concept rooted in classical genetics[11], describes cell death resulting from the combination of a genetic variant and a chemical perturbation, where each individual perturbation is viable. By exploiting genetic variants (e.g., somatic mutations, copy number variations, chromosomal rearrangements, or gene expression changes) that differentiate tumor from normal tissue, synthetic lethal interactions provide a therapeutic window for selectively targeting cancer cells. The potential of synthetic lethality is best exemplified by the development and FDA approval of PARP inhibitors for patients with BRCA-mutated ovarian, breast, and prostate cancers[12,13]. There is also value in identifying synthetic rescue interactions, where a cytotoxic compound has reduced efficacy in the presence of a particular genetic variant, thus providing insights into drug resistance mechanisms[14]. The only existing strategies for identifying clinically relevant chemical–genetic interactions for more than a handful of genetic variants and compounds rely entirely on predictive approaches. Some of these predictions are based on genetic or chemical–genetic interactions identified in yeast[15,16], while others, such as the Cancer Cell Line Encyclopedia[17], Genomics of Drug Sensitivity in Cancer[18–20], Cancer Therapeutics Response Portal[21–23], and PRISM[24,25], are based on computational methods that correlate genetic and molecular features of human cancer cell lines with drug response. While these correlative approaches have been useful, they are limited by the fact that many features are rare and lack sufficient representation—or are not even present—in current cancer cell line collections, reducing the statistical power to detect significant correlations. Related to this, even correlations involving more common features are confounded by the multitude of additional features that also distinguish each cell line. Thus, these approaches still require direct experimental validation, ideally in a manner that tests individual features along with their corresponding isogenic controls in relevant mammalian models.

Here, to systematically and directly measure the contribution of individual genes to acute drug response, we devise quantitative and multiplexed analysis of phenotype by sequencing (QMAP-Seq). Unlike most chemical–genetic strategies in mammalian systems, QMAP-Seq is characterized by short-term compound treatment, which better recapitulates the timing of most high-throughput drug screening assays and enables testing of thousands of compounds in parallel. As proof-of-concept, we apply QMAP-Seq to the protein homeostasis (proteostasis) network, a critical set of cellular stress response factors that maintain proper protein function from synthesis to folding to degradation. Because individual proteostasis factors are activated to varying degrees across tumors to cope with cancer-associated and drug-induced stress[26–31], but are not easily druggable[32], we reasoned that QMAP-Seq could be used to identify synthetic lethal chemical–genetic interactions dependent on the activation status of these factors. Furthermore, because the individual branches of the proteostasis network are typically studied in isolation, and the functional relationship within and between the branches remains largely unexplored, we postulated that QMAP-Seq could also be used to provide functional insight into the proteostasis network. We first perform QMAP-Seq with one cell line and demonstrate that it generates precise and accurate quantitative measures of compound efficacy that are concordant with established cell viability assays based on live-cell imaging. We then expand QMAP-Seq to multiple cell lines, which enables the parallel measurement of 86,400 cell viability phenotypes in a single experiment. Altogether, we identify 60 sensitivity interactions and 124 resistance interactions and validate a subset of these interactions individually using an established metabolic-based cell viability assay. This work illustrates the power of systematic, high-throughput chemical–genetic profiling in mammalian systems.

## Results

**Engineering barcoded breast cancer cell lines with inducible single-gene knockouts.** Because the proteostasis network is heterogeneously activated in cancer, but is not easily druggable and is incompletely understood, we created a custom sgRNA library to disrupt a set of 10 genes that play pivotal roles in regulating the proteostasis network. These genes included critical factors involved in the heat-shock response (HSF1, HSF2), unfolded protein response (IRE1 (*ERN1*), XBP1, ATF3, ATF4, ATF6), oxidative stress response (NRF2 (*NFE2L2*), KEAP1), and in autophagy (ATG7). As a control to validate that QMAP-Seq can detect known chemical–genetic interactions, our library also contained an sgRNA targeting SLC35F2, a solute carrier required for cellular uptake of the cytotoxic compound, YM155[33]. We engineered MDA-MB-231 triple-negative breast cancer cells with these 11 single-gene knockouts and a pool of five non-targeting (NT) sgRNA controls. Because constitutive expression of Cas9 can result in off-target effects[34] and cell toxicity[35], we designed a system with doxycycline-inducible Cas9, providing temporal control over gene knockout (Supplementary Fig. 1a). Cas9 was induced in a doxycycline dose-dependent manner (Supplementary Fig. 1b). To enable future pooling and identification of multiple cell lines, we introduced unique 8 bp cell line barcode sequences downstream of the sgRNA within the lentiGuide-Puro plasmid (Supplementary Fig. 1c).

To assess the efficacy of the sgRNAs, we performed Western blot analysis 96 h after Cas9 induction and confirmed efficient whole population knockout of the proteostasis factors (Supplementary Fig. 1d). Although we could not easily detect KEAP1 expression or knockout by western blot, we observed the expected upregulation of NRF2 protein levels in the KEAP1 knockout cells (Supplementary Fig. 1d). In addition, we used an ATP-based cell

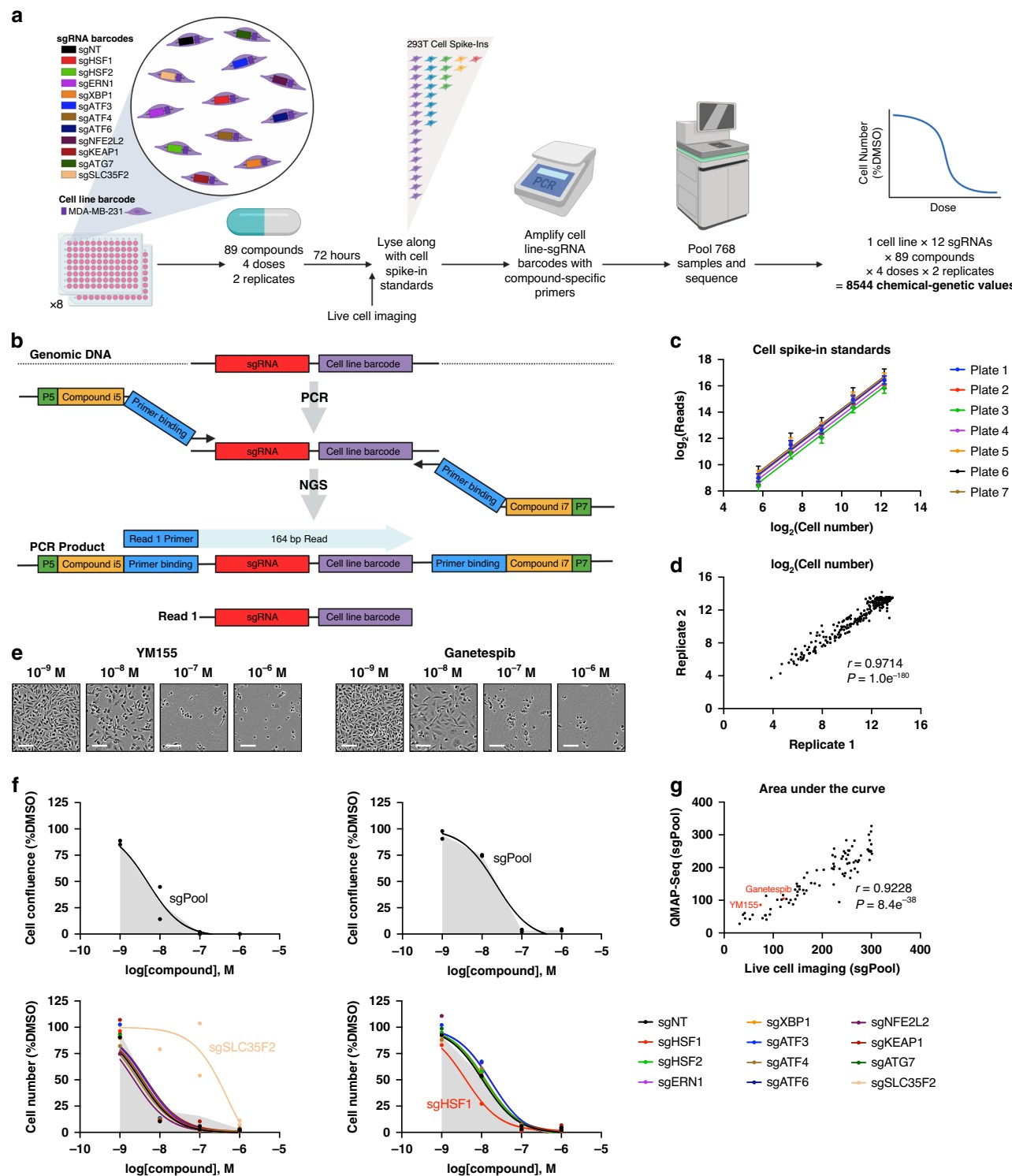

viability assay to confirm the ability of the sgRNAs that target SLC35F2 to confer resistance to YM155 (Supplementary Fig. 1e).

**QMAP-Seq generates precise and accurate quantitative measures of drug response**. We sought to develop and apply QMAP-Seq to quantify the response of a mixed pool of MDA-MB-231 cells possessing our panel of proteostasis factor knockouts to treatment with 89 compounds targeting diverse biological processes at four doses in duplicate (Fig. 1a). Our experimental workflow involved inducing Cas9 to initiate knockout, treating with either DMSO control or compound for 72 h, and then

preparing crude cell lysates. Previous studies have demonstrated the utility of spike-in standards for quantification when performing RNA-Seq[36] and ChIP-Seq[37,38]. To enable a quantitative assay, we introduced 293T cell spike-in standards composed of predetermined numbers of cells for each of five unique sgNT barcodes into each sample. Spike-in cell numbers were customized for each experiment to cover the expected range of cell numbers for any individual perturbation at the time of cell lysis (see "Methods" for details). We next amplified the 768 samples, corresponding to distinct compound-dose-replicate combinations, using unique sets of i5 and i7 indexed primers (Fig. 1b,

**Fig. 1 QMAP-Seq generates precise and accurate quantitative measures of drug response. a** Experimental workflow for QMAP-Seq with one cell line. **b** Schematic of QMAP-Seq library preparation using unique sets of i5/i7 indexed primers followed by next-generation sequencing of amplicons. **c** Standard curves generated from five uniquely barcoded 293T cell spike-in standards introduced at known cell numbers. Data are represented as mean number of sequencing reads across the six or eight DMSO samples on a plate ± standard deviation. **d** Scatterplot of interpolated cell number for two biologically independent replicates. Statistical significance of Pearson correlation was determined using a two-tailed test ($n = 288$ compound-dose combinations). **e** Live-cell imaging of MDA-MB-231 sgPool cells 72 h after treatment with YM155 or Ganetespib. Images are representative of two biologically independent replicates. Scale bar = 100 μm. Source data are provided as a Source data file. **f** Top: Dose–response curves for MDA-MB-231 sgPool cells as measured using live-cell imaging 72 h after treatment with YM155 or Ganetespib. Bottom: Dose–response curves for 12 genetic perturbations of MDA-MB-231 cells as measured using QMAP-Seq 72 h after treatment with YM155 or Ganetespib. Each data point represents one of two biologically independent replicates. The shaded region indicates the area under the curve (AUC) for sgPool. **g** Scatterplot of the dose–response curve AUC for sgPool as determined using live-cell imaging versus QMAP-Seq. Statistical significance of Pearson correlation was determined using a two-tailed test ($n = 89$ compounds). Source data are available in the Source data file.

Supplementary Data 1, Supplementary Data 2). To facilitate Illumina sequencing, our PCR primers incorporated P5 and P7 adaptors complementary to flow cell oligos. To improve sequence diversity, we utilized a mix of P5 primers with varying stagger lengths. After PCR amplification, we pooled and purified the PCR products followed by Illumina sequencing with a single 164 bp read to sequence the sgRNA and cell line barcodes (Fig. 1b).

To simplify the processing of large numbers of samples, we built a multistep QMAP-Seq bioinformatic analysis pipeline (Supplementary Fig. 2). First, the pipeline demultiplexed the 768 individual samples according to i5 and i7 index sequences (Supplementary Data 3 and 4). Second, it extracted the cell line barcode (Supplementary Data 5) and sgRNA barcode (Supplementary Data 6) from each read and counted the number of reads for each cell line-sgRNA pair. Third, the pipeline used the cell spike-in standards to generate a sample-specific standard curve and used the standard curve to interpolate cell number from sequencing reads. Finally, it calculated the number of cells for each cell line-sgRNA pair in the presence of compound relative to DMSO control.

We next performed a series of analyses to assess the quality of the data generated from QMAP-Seq. We first asked whether we could resolve differences in cell number by next-generation sequencing. Indeed, analysis of the cell spike-in standards revealed the expected increase in sequencing reads with increased input cell number (Fig. 1c). To assess the precision of QMAP-Seq, we compared the interpolated cell number between the two replicates for every compound-dose pair. Importantly, QMAP-Seq replicates were highly correlated ($r = 0.9714$) (Fig. 1d). To compare QMAP-Seq with an established method of measuring cellular response to compounds, we assessed the percent confluence of the population of cells in each compound-treated well using live-cell imaging immediately prior to cell lysis (Fig. 1a). As expected, live-cell imaging revealed a reduction in cell confluence with increasing concentrations of YM155 (Fig. 1e, f, top left). The area under the dose–response curve (AUC) for the population of cells as determined using QMAP-Seq was similar to live-cell imaging analysis (live-cell imaging AUC = 74.12, QMAP-Seq AUC = 85.95). Importantly, QMAP-Seq was further able to resolve differences in drug response between the knockouts, such as knockout of SLC35F2 conferring resistance to YM155 (Fig. 1f, bottom left). As another example, live-cell imaging revealed a dose-dependent reduction in cell confluence with the HSP90 inhibitor, Ganetespib (Fig. 1e, f, top right). Once again, the AUC for the population of cells calculated using QMAP-Seq was similar to live-cell imaging analysis (live-cell imaging AUC = 120.90, QMAP-Seq AUC = 104.80). Furthermore, QMAP-Seq detected that knockout of HSF1 sensitized cells to Ganetespib, confirming another previously established chemical–genetic interaction[39] (Fig. 1f, bottom right). Notably, the nearly identical results obtained from the two assays were not

limited to these two compounds. The AUCs for the population of cells calculated using QMAP-Seq were remarkably concordant with those calculated using live-cell imaging across all 89 compounds ($r = 0.9228$), demonstrating the high degree of accuracy of QMAP-Seq (Fig. 1g). Taken together, these results indicate that QMAP-Seq generates precise, accurate, and sensitive quantitative measures of pharmacologic response in pooled format.

**Expanding QMAP-Seq to multiple cell lines.** Because the genetic and epigenetic background of a cell line can impact therapeutic response, we next expanded QMAP-Seq to multiple cell lines (Fig. 2a). We selected a panel of breast cancer cell lines comprising three major subtypes of breast cancer: ER+ (ZR-75-1), HER2+ (SKBR3), and triple-negative (HCC-38, MDA-MB-231, BT-20). We assembled a 180 compound collection for QMAP-Seq (Supplementary Fig. 3, Supplementary Data 7) in a manner that allowed us to achieve two primary objectives for this assay. First, we selected FDA-approved drugs or compounds in clinical trials to facilitate the discovery of clinically relevant chemical–genetic interactions and to enable drug repurposing (Fig. 2b). This included chemotherapeutics and targeted therapies that are either standard-of-care or currently being investigated in the context of breast cancer. Second, we selected compounds that target biological processes from 19 diverse pathways because proteostasis factors broadly impact cell biology (Fig. 2c).

To ensure that each cell line would be similarly represented in our assay despite different doubling times, we measured the relative abundance of each of the five cell lines grown in heterogeneous pools. We prepared five pools, each containing 20% of one cell line expressing ZsGreen and 20% of each of the other four cell lines expressing dTomato (Fig. 2d). We cocultured and analyzed the pools by flow cytometry to quantify the percentage of GFP positive cells over time. We found that SKBR3 cells were the most depleted, whereas MDA-MB-231 cells were the most enriched (Fig. 2e, left). We used the relative cell abundances from this competition experiment to mathematically model an optimized pooling ratio to ensure adequate representation of all cell lines at seven days post-pooling, the timepoint when the cell lines are exposed to compounds during QMAP-Seq (Fig. 2e, right, 0 days). Optimized pools predicted to contain 20% of each cell line after 7 days exhibited similar representation of the cell lines at this timepoint (Fig. 2e, right, 7 days).

After pooling the five cell lines each possessing our panel of proteostasis factor knockouts at the optimized pooling ratio, we performed the QMAP-Seq workflow (Fig. 2a). Seven days after thawing, we treated the pooled cells with compounds or DMSO. After 3 days of treatment, we measured the representation of the cell line-sgRNA pairs in the DMSO samples. Each of the cell lines was covered by at least 10% of the total sequencing reads, indicating adequate representation of the five cell lines (Fig. 2f).

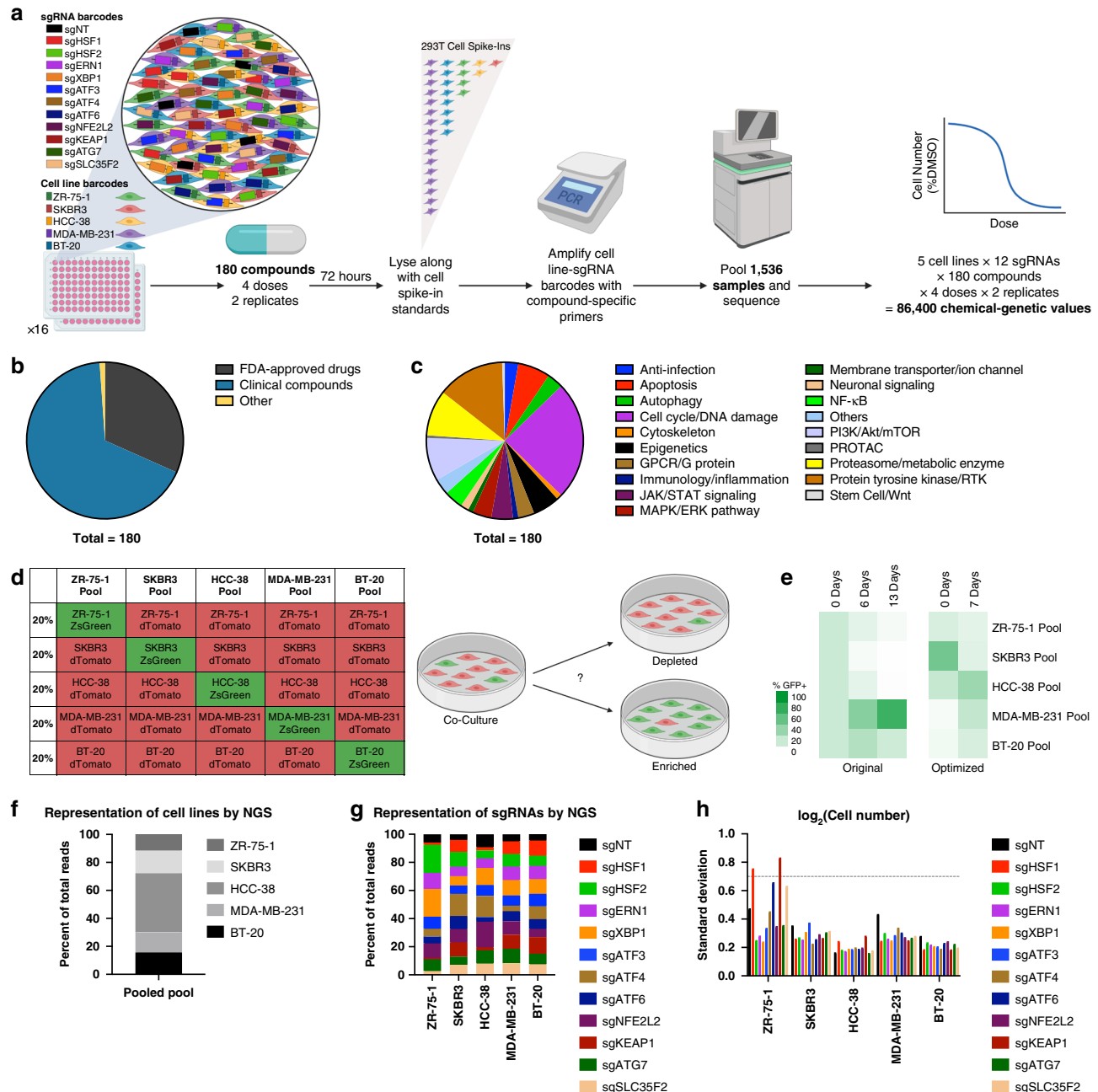

**Fig. 2 Expanding QMAP-Seq to multiple cell lines. a** Experimental workflow for QMAP-Seq with five cell lines. **b** Pie chart showing breakdown of 180 compounds by stage of development. **c** Pie chart showing breakdown of 180 compounds by pathway. **d** Schematic of competition experiment to optimize the starting representation of the five co-cultured breast cancer cell lines. Original cell line pools were prepared by mixing equal numbers of one cell line expressing ZsGreen with each of the other four cell lines expressing dTomato. Flow cytometry analysis measured the percentage of GFP positive cells in each pool over time. **e** Heat maps displaying the percentage of GFP positive cells at various time points as measured using flow cytometry. Left: Original cell line pools that started with 20% of each cell line on Day 0. Right: Optimized cell line pools predicted to contain 20% of each cell line on Day 7. Source data and gating strategy are provided as a Source data file. **f** Representation of five breast cancer cell lines as measured by counting the number of sequencing reads for each cell line barcode across the 96 DMSO samples. **g** Representation of sgRNAs as measured by counting the number of sequencing reads for each sgRNA barcode relative to the total number of sequencing reads for that cell line across the 96 DMSO samples. **h** Standard deviation of the interpolated cell number for each cell line-sgRNA pair across the 96 DMSO samples. Dotted line indicates threshold for excluding cell line-sgRNA pairs with high variability. Source data are available in the Source data file.

We also observed sufficient representation of most cell line-sgRNA pairs (Fig. 2g). Unsurprisingly, the two cell line-sgRNA pairs with the lowest representation in the pool (ZR-75-1 sgHSF1 and ZR-75-1 sgKEAP1) exhibited the greatest variation between DMSO samples (standard deviation >0.7) and were thus excluded from downstream analysis (Fig. 2h).

Next, we asked whether cell spike-in standards could reduce the technical variation between plates that we observed in the raw sequencing reads across the DMSO control samples (Supplementary Fig. 4b, top). For example, Plate 6 had markedly fewer sequencing reads compared to the other plates, but this plate also had a standard curve with a lower y-intercept, reflecting

proportionally lower total reads compared to the other plates (Supplementary Fig. 4a). Thus, utilizing sample-specific standard curves to interpolate cell number from sequencing reads improved this and other variation between plates (Supplementary Fig. 4b, bottom). For each of the five cell lines, we observed a statistically significant decrease in variation between plates upon interpolation (ZR-75-1: SD, −53% compared with raw reads, F test: $P = 6.3e^{−3}$) (SKBR3: SD, −66% compared with raw reads, F test: $P = 1.4e^{−4}$) (HCC-38: SD, −88% compared with raw reads, F test: $P = 5.9e^{−11}$) (MDA-MB-231: SD, −71% compared with raw reads, F test: $P = 2.4e^{−5}$) (BT-20: SD, −67% compared with raw reads, F test: $P = 1.1e^{−4}$) (Supplementary Fig. 4c). Thus, cell spike-in standards serve as an internal control for normalizing data for each sample, thereby reducing technical variation and improving overall data quality.

**Identification and validation of cell line–gene–drug interactions**. We next examined whether we could detect known cell line-specific and gene-specific drug vulnerabilities within complex mixtures of cells (five cell lines × 12 genetic perturbations) using QMAP-Seq. As expected, the ER+ cell line ZR-75-1 was preferentially sensitive to 4-Hydroxytamoxifen, and the HER2+ cell line SKBR3 was preferentially sensitive to Lapatinib (Fig. 3a). In addition, knockout of SLC35F2 conferred resistance to YM155, most prominently in the triple-negative breast cancer cell lines that were most sensitive to this compound (Fig. 3a).

In total, a single next-generation sequencing run consisting of 1.3 billion reads provided the capacity to make 86,400 relative cell number measurements (Supplementary Data 8), plot 10,800 dose–response curves (Supplementary Fig. 5), and calculate 21,600 AUCs (Supplementary Data 9). Despite profiling five times as many cell lines for this QMAP-Seq experiment compared to our pilot QMAP-Seq experiment with one cell line, the AUC measurements from the common cell line-compound pairs were highly correlated between these two independent experiments ($r = 0.9324$) (Fig. 3b). Thus, neither mixing different parental cell lines nor increasing the total number of cell types analyzed altered the performance of QMAP-Seq.

Statistical analysis revealed 60 cell line–gene–drug interactions that conferred compound sensitivity (AUC difference < −25 and $P < 0.05$) and 124 cell line–gene–drug interactions that conferred compound resistance (AUC difference >25 and $P < 0.05$) compared to sgNT (see "Methods" for details) (Fig. 3c, Supplementary Data 10). To characterize the type of pathways enriched among the compounds involved in the top sensitivity and resistance interactions, we compared the expected distribution of the 19 pathways in our compound collection (Fig. 2c) with the observed distribution of these pathways among the 60 sensitivity interactions and the 124 resistance interactions. Notably, the most significantly enriched pathway among the sensitivity interactions was proteasome and metabolic enzyme compounds, particularly proteasome inhibitors (Fig. 3d), suggesting that proteasome inhibition is especially lethal under conditions of proteostasis factor depletion. Epigenetics and apoptosis were also significantly enriched among the sensitivity interactions, whereas anti-infection, cytoskeleton, protein tyrosine kinase, cell cycle, and DNA damage were the most enriched pathways among the resistance interactions (Fig. 3d).

Among the top chemical–genetic interactions identified using QMAP-Seq (Fig. 3e) was loss of four proteostasis factors (ATF4, HSF2, HSF1, NFE2L2) further sensitizing MDA-MB-231 cells to the proteasome inhibitor, Carfilzomib (Fig. 3f, top), corroborating previous findings that loss of these factors enhances sensitivity to proteasome inhibitors in cancer cells[40–43]. To validate these pooled screening hits in traditional arrayed format, we treated

MDA-MB-231 cells possessing individual gene knockouts with Carfilzomib for 72 h and measured their intracellular reducing potential as a proxy of cell viability. As we observed using QMAP-Seq, knockout of these proteostasis factors further sensitized MDA-MB-231 cells to Carfilzomib (Fig. 3f, bottom). We conclude that proteostasis factor depletion is synthetic lethal with proteasome inhibition and more broadly, chemical–genetic interactions uncovered using heterogenous mixtures of cell lines are reproducible using homogenous cell lines.

**QMAP-Seq enables proteostasis network mapping in breast cancer**. To further investigate the strongest chemical–genetic interactions in our dataset, we assembled a chemical–genetic interaction map of the highest confidence interactions (absolute AUC difference >60 and $P < 0.05$) (Fig. 4a). This network incorporated both synthetic lethal gene–drug interactions and synthetic rescue gene–drug interactions. The map revealed hub compounds that synergize with loss of multiple proteostasis factors. For example, the proteasome inhibitor, Carfilzomib displayed five synthetic lethal chemical–genetic interactions with proteostasis factors from distinct branches of the proteostasis network. We also observed hub characteristics for compounds not previously connected to proteostasis, including the procaspase-3 activator, PAC1, and the DNA methyltransferase inhibitor, Lomeguatrib.

To model the overall structure of the proteostasis network, we assessed the functional similarity of each proteostasis factor's chemical–genetic interaction profiles. Specifically, we quantified the Spearman correlation between all gene–gene pairs based on AUC difference across all cell line and compound contexts (Fig. 4b). Using this approach, genes with similar compound sensitivity and resistance profiles clustered together, revealing several known genetic relationships (Fig. 4c). For example, because KEAP1 negatively regulates NRF2 (*NFE2L2*), the master transcriptional regulator of the oxidative stress response[44], we expected a low degree of correlation between these two factors. Indeed, NFE2L2 and KEAP1 displayed the third lowest correlation. In addition, we observed a high correlation between ERN1 and XBP1, which was expected given that the ER stress sensor, IRE1 (*ERN1*), activates XBP1 through mRNA splicing[45].

This approach also revealed previously unknown genetic relationships (Fig. 4c). For example, the highest correlation was observed between NFE2L2 and XBP1. While the relationship between these two factors in the context of breast cancer is unknown, XBP1 has been shown to activate NRF2 in atherosclerosis[46] and retinal pigment epithelium cells[47]. Furthermore, the paralogs HSF1 and HSF2 have been reported to function cooperatively[48–51], antagonistically[52], and in a context-dependent manner[53]. Interestingly, HSF1 and HSF2 displayed the second highest correlation, providing evidence in support of a cooperative interaction in breast cancer. The lowest correlation genetic interaction was observed between ATF4 and KEAP1, which is not entirely surprising considering KEAP1 has been reported to negatively regulate ATF4 expression in other cancer models[54]. Altogether, we conclude that chemical–genetic profiling using QMAP-Seq provides insight into the organization of the proteostasis network in breast cancer cells and has the power to reveal genetic relationships.

**Discussion**
Here, we introduce QMAP-Seq, a highly multiplexed chemical–genetic profiling strategy that enables systematic phenotyping of dozens of cell lines with defined genetic perturbations across thousands of individual compound treatments. We also present a bioinformatic analysis pipeline that simplifies the processing of

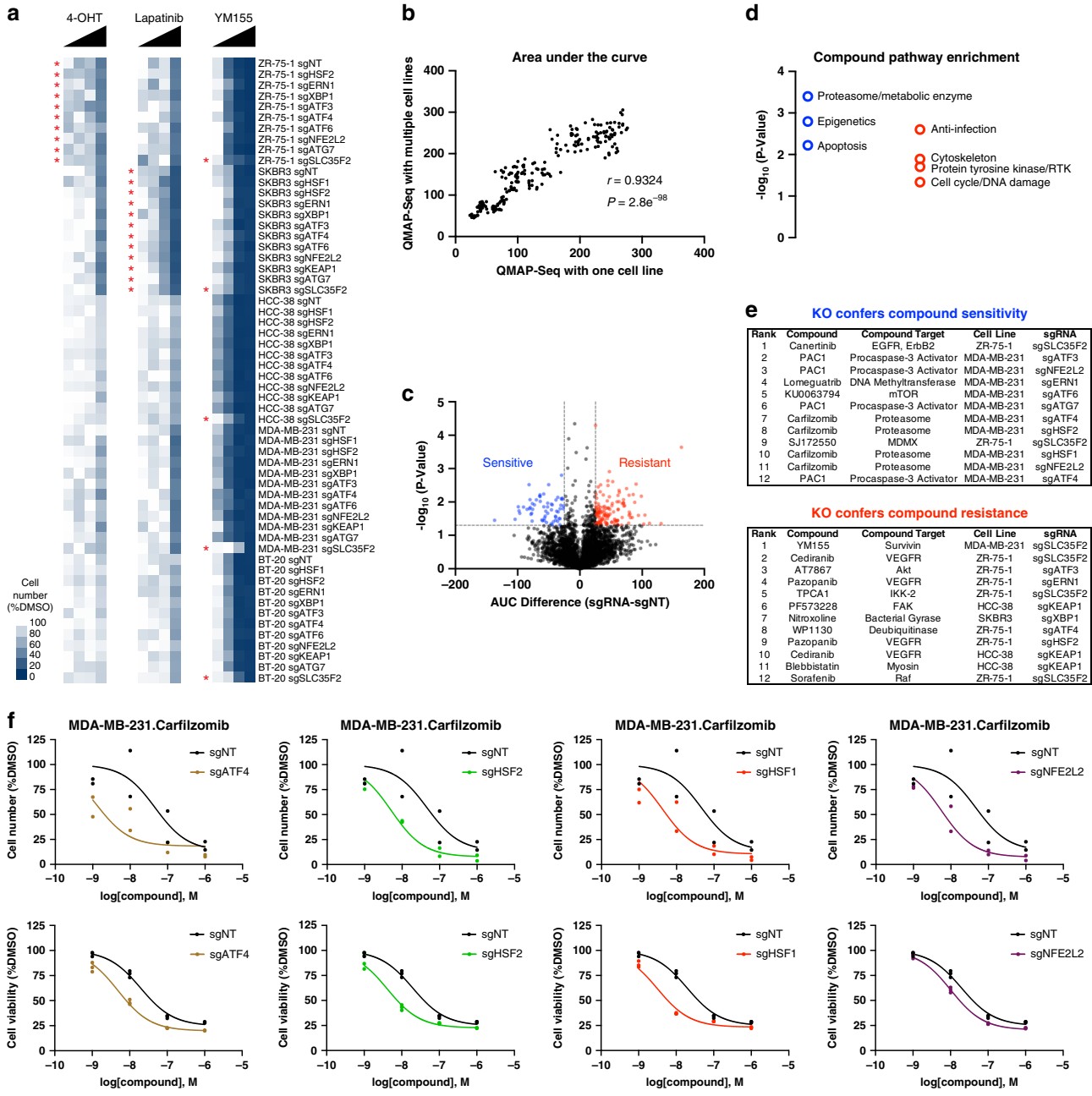

**Fig. 3 Identification and validation of cell line–gene–drug interactions. a** Heat maps displaying the relative cell number for each cell line-sgRNA pair 72 h after treatment with 4-Hydroxytamoxifen (4-OHT), Lapatinib, or YM155 as measured using QMAP-Seq. Data are represented as mean of two biologically independent replicates. Asterisks denote positive controls. **b** Scatterplot of the dose–response curve AUC as determined using QMAP-Seq with one cell line versus QMAP-Seq with multiple cell lines. Common compounds for MDA-MB-231 cells are shown. Statistical significance of Pearson correlation was determined using a two-tailed test ($n = 220$ compound-sgRNA combinations). **c** Volcano plot depicting cell line–gene–drug interactions. Magnitude was determined by calculating the difference in mean AUC between sgRNA and sgNT for every cell line-compound combination. Statistical significance was determined using an unpaired, two-tailed $t$ test ($n = 2$ biologically independent replicates). Blue dots indicate interactions where the knockout confers greater sensitivity than sgNT. Red dots indicate interactions where the knockout confers greater resistance than sgNT. **d** Pathways targeted by compounds involved in the top 60 sensitivity interactions (blue circles) or top 124 resistance interactions (red circles) that were significantly enriched compared to expected pathway representation. Statistical significance of pathway enrichment was determined using a one-tailed binomial test to compare observed distribution with expected distribution ($n = 180$ compounds). **e** List of the top 12 conditions that confer compound sensitivity or resistance. **f** Top: Dose–response curves for four of the top chemical–genetic interactions as measured using QMAP-Seq. Each data point represents one of two biologically independent replicates. Bottom: Dose–response curves as measured using Resazurin Cell Viability Kit. Each data point represents one of three biologically independent replicates. For clarity, individual proteostasis factor knockout curves are partioned across four panels; sgNT is same in all cases.

thousands of fastq files into over 10,000 dose–response curves. QMAP-Seq has several major advantages over existing methods for the identification of chemical–genetic interactions in mammalian cells. For one, our pooled approach provides substantial

gains in throughput, facilitating over 50 times as many cell viability measurements per sample as established arrayed assays. By introducing sample-specific indexes, thousands of compound treatment samples are further pooled and sequenced together in a

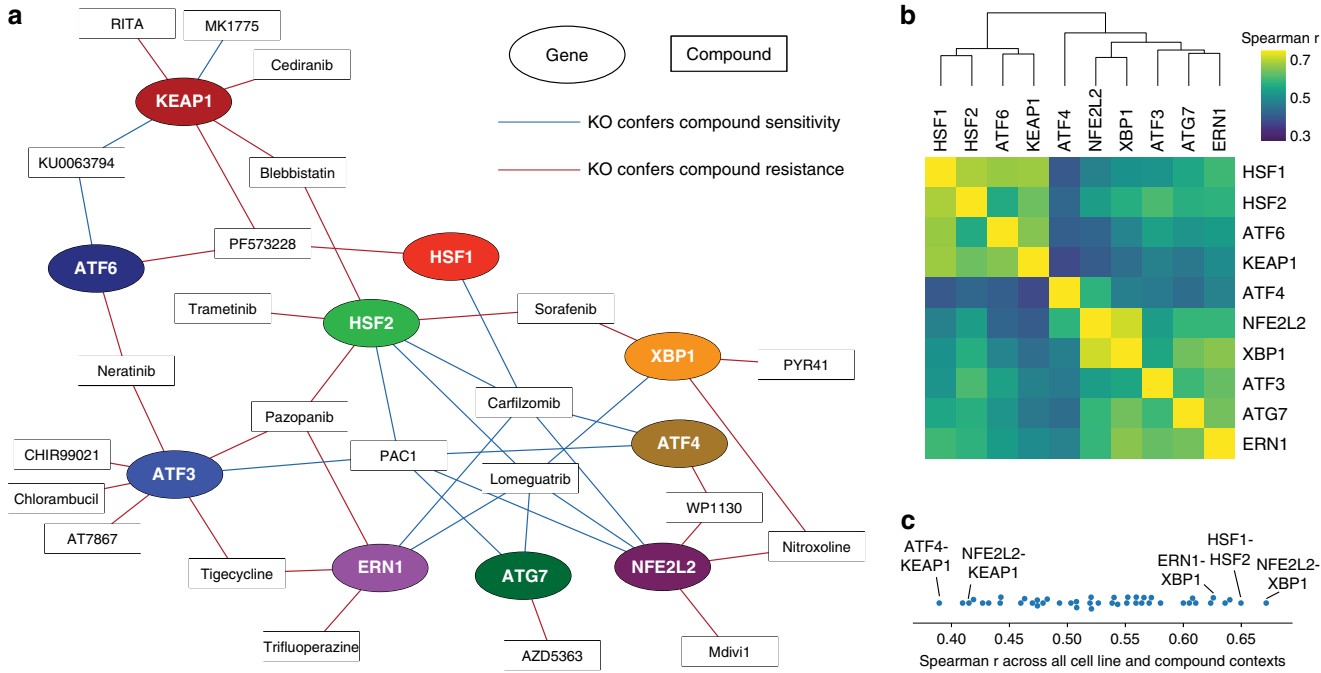

**Fig. 4 QMAP-Seq enables proteostasis network mapping in breast cancer. a** Chemical–genetic interaction map displaying the highest confidence interactions identified by QMAP-Seq (absolute AUC difference >60 and *P* < 0.05). Genes are represented as oval nodes, and compounds are represented as rectangular nodes. Chemical–genetic interactions where the knockout confers compound sensitivity (i.e., synthetic lethal interactions) are represented as blue edges. Chemical–genetic interactions where the knockout confers compound resistance (i.e., synthetic rescue interactions) are represented as red edges. **b** Spearman correlation matrix of gene–gene correlations based on AUC differences across all cell lines and compound contexts (*n* = 488 cell line-compound combinations). **c** Distribution of gene–gene Spearman correlations across all cell line and compound contexts. Known and previously unknown genetic relationships are labeled.

single next-generation sequencing run. This distinguishes QMAP-Seq from PRISM[24], an approach that uses microsphere bead technology for multiplexing cell lines, but not compound treatments. Another important feature of QMAP-Seq is the use of genetically engineered isogenic cell line pairs, which enables direct assessment of the contribution of a single genetic perturbation to compound sensitivity. It is worth noting, however, that other pooled assays could in principle profile isogenic cell line pairs, while QMAP-Seq could be performed with pools of non-isogenic cell lines. QMAP-Seq leverages readily available next-generation sequencing, which itself continues to improve in throughput, and is compatible with common sequencing libraries, such as RNA-Seq. As a result, the costs associated with QMAP-Seq are 1/10th the cost of an ATP-based cell viability assay of the same magnitude performed in 384-well format. The compound requirements are also lower, which is particularly useful when profiling scarce compounds or natural products.

Importantly, the short-term nature of QMAP-Seq (i.e., 72 h compound treatment) enables profiling of thousands of compounds, which would be technically challenging with existing genome-scale chemical–genetic methods that require several weeks of passaging and treating cells prior to readout. The short-term format also minimizes potential secondary effects that may arise due to several weeks of exposure to compounds, and thus identifies the most relevant chemical–genetic interactions. Collectively, these attributes of QMAP-Seq enable accurate mapping of biological network structure, as we demonstrated for the proteostasis network.

Nevertheless, QMAP-Seq has certain limitations. For one, pooling multiple cell lines requires optimization of cell pooling ratios to ensure adequate representation. However, this could be bypassed by pooling cell lines with similar doubling times[25] or by profiling one cell line at a time. Additionally, factors secreted by

one cell line could conceivably alter the compound sensitivity of another cell line in the pool. Although such paracrine-mediated effects would certainly warrant further studies, we (Fig. 3b, f) and others[24,25] have yet to observe any major difference in drug response measurements from pooled versus arrayed formats. Finally, while QMAP-Seq is compatible with standard cell viability readouts, it is less suitable for assessing non-cell viability phenotypes. It could, however, be adapted to non-standard readouts by isolating cells that possess the desired phenotype prior to sequencing.

We envision several potential applications of QMAP-Seq. For one, networks of genes identified using genome-scale chemical–genetic studies could be investigated further using QMAP-Seq with a broader range of compounds and a more focused set of genes. Notably, the QMAP-Seq experiments described in this paper were performed without significant automation. By employing automation, this assay could readily be scaled to encompass an even larger number of compounds. In addition, QMAP-Seq could be used to directly characterize the impact of individual cancer-specific alterations[55,56] on therapeutic response, which may improve patient stratification and treatment outcomes. Moreover, QMAP-Seq could be adapted to enable more sophisticated assays. For example, to better model the contribution of the tumor microenvironment to drug response, it could be used for chemical–genetic profiling of barcoded cancer cells grown in the presence of stromal cells. Lastly, other types of perturbations besides chemical treatments could be applied, such as nutrient conditions, to assess the consequence of diverse perturbations on cell viability across multiple genetic contexts. As more QMAP-Seq data is collected over time from these and other potential applications, we see value in building searchable databases that could serve as a resource for the broader scientific community.

Overall, we report the development of the QMAP-Seq experimental and bioinformatic pipeline and the application of this chemical–genetic profiling approach to the proteostasis network. QMAP-Seq addresses the need for a direct, non-correlative method for assessing compound selectivity across multiple cellular and genetic contexts in a high-throughput and scalable manner. It provides a path toward cancer precision medicine by predicting clinically actionable synthetic lethal and synthetic rescue interactions. QMAP-Seq represents the first application of chemical–genetic profiling to map biological networks and reveal functional genetic relationships in mammalian cells. Above all, the QMAP-Seq platform is well-suited to answer a broad range of clinical and biological questions and can be readily adopted by standard laboratories without the need for highly specialized equipment.

## Methods

**Plasmid construction.** For cloning doxycycline-inducible Cas9, 3xFLAG-Cas9-EGFP was amplified from pSpCas9(BB)-2A-GFP (Addgene, Plasmid #48138)[57] with Cas9 EcoRI F and EGFP BamHI R (Supplementary Data 11) and inserted within the EcoRI and BamHI restriction sites of the pLVX-TetOne vector (Clontech, #631846) using the In-Fusion HD Cloning Kit (Clontech, #638909), according to manufacturer's instructions.

For inserting cell line barcodes downstream of the gRNA scaffold and upstream of cPPT/CTS within the lentiGuide-Puro vector (Addgene, Plasmid #52963)[58], EcoRI and BamHI restriction sites were introduced flanking the future cell line barcode insertion site by amplifying two overlapping fragments with Barcode Frag1 F/R and Barcode Frag2 F/R (Supplementary Data 11). The two fragments were inserted within the Mph1103I and SmaI restriction sites of lentiGuide-Puro using the In-Fusion HD Cloning Kit (Clontech, #638909), according to the manufacturer's instructions. Six unique 8 bp cell line barcode sequences were inserted within the EcoRI and BamHI restriction sites of the modified lentiGuide-Puro vector by resuspending the top and bottom strands of the cell line barcode oligos (Supplementary Data 11) to a final concentration of 100 μM. Oligo pairs were phosphorylated and annealed by combining 1 μL 100 μM cell line barcode top, 1 μL 100 μM cell line barcode bottom, 5 μL 2x Quick Ligation Buffer, 1 μL T4 PNK, and 2 μL ddH$_2$O and incubating in thermocycler as follows: 37 °C for 30 min, 95 °C for 5 min, ramp down to 25 °C (5 °C per minute). Cell line barcode oligos were inserted within the EcoRI and BamHI restriction sites of the modified lentiGuide-Puro vector.

Gene-targeting sgRNAs sourced from the Brunello Human CRISPR Knockout Pooled Library[59] and non-targeting (NT) sgRNAs sourced from ref. [58] and ref. [9] were inserted into the cell line barcoded lentiGuide-Puro vectors by resuspending the top and bottom strands of the sgRNA oligos (Supplementary Data 11) to a final concentration of 100 μM. Oligo pairs were phosphorylated and annealed by combining 1 μL 100 μM sgRNA top, 1 μL 100 μM sgRNA bottom, 5 μL 2x Quick Ligation Buffer, 1 μL T4 PNK, and 2 μL ddH$_2$O and incubating in thermocycler as follows: 37 °C for 30 min, 95 °C for 5 min, ramp down to 25 °C (5 °C per minute). sgRNA oligos were cloned into the barcoded lentiGuide-Puro vectors by combining 100 ng vector, 2 μL 1:200 diluted oligo duplex, 2 μL 10x FastDigest Buffer, 1 μL 10 mM DTT, 1 μL 10 mM ATP, 1 μL BsmBI, 0.5 μL Quick Ligase, and ddH$_2$O to a final volume of 20 μL and incubating in thermocycler as follows: 37 °C for 5 min, 21 °C for 5 min, repeat for a total of 6 cycles. Ligation reactions were treated with PlasmidSafe exonuclease by combining 11 μL ligation reaction, 1.5 μL 10x PlasmidSafe Buffer, 1.5 μL 10 mM ATP, and 1 μL PlasmidSafe exonuclease and incubating in thermocycler as follows: 37 °C for 30 min, 70 °C for 30 min. Ligation reactions were transformed into Stbl3 cells. All plasmid sequences were verified by Sanger sequencing.

**Cell culture.** 293T, ZR-75-1, SKBR3, HCC-38, MDA-MB-231, and BT-20 cells were obtained from ATCC. 293T cells were cultured in DMEM medium (Gibco, #11995073) supplemented with 10% Tet System Approved Fetal Bovine Serum (Clontech, #631106) and 1% Penicillin/Streptomycin (Gibco, #15140122). To facilitate co-culturing cell lines in mixtures, ZR-75-1, SKBR3, HCC-38, MDA-MB-231, and BT-20 cells were cultured individually or collectively in a common growth medium: RPMI-1640 medium (Gibco, #11875119) supplemented with 10% Tet System Approved Fetal Bovine Serum (Clontech, #631106) and 1% Penicillin/Streptomycin (Gibco, #15140122). All cell lines were authenticated at the University of Arizona Genetics Core and tested negative for mycoplasma contamination.

**Cell engineering.** For engineering Cas9-expressing breast cancer cell lines, virus was produced from pLVX-TetOne Cas9 lentiviral transfer plasmid as specified in the application note for Lipofectamine 3000 Transfection Reagent (Invitrogen, #L3000015), but substituting pMD2.G envelope plasmid and psPAX2 packaging plasmid for ViraPower Lentiviral Packaging Mix. ZR-75-1, SKBR3, HCC-38,

MDA-MB-231, and BT-20 cells were transduced with virus and induced with either 100 ng/mL doxycycline (ZR-75-1, SKBR3, BT-20) or 10 ng/mL doxycycline (HCC-38, MDA-MB-231) for 7 days (refreshing doxycycline every 3 days). Induced cells were gated into three equal-size bins based on GFP brightness and sorted for medium GFP+ cells.

For introducing sgRNA libraries, pooled sgRNA plasmid libraries possessing appropriate cell line barcodes were generated by combining equal amounts of 11 gene-targeting sgRNA plasmids and a pool of five NT sgRNA plasmids (sgNT_0-4). Virus was produced with Lipofectamine 3000 Transfection Reagent (Invitrogen, #L3000015) as described above and was functionally titered on its respective cell line using serially-diluted virus. ZR-75-1, SKBR3, HCC-38, MDA-MB-231, and BT-20 Cas9 cells were transduced with appropriate sgRNA libraries at MOI = 0.3, maintaining coverage of at least 7500 cells per sgRNA. Transduced cells were selected using 2 μg/mL puromycin (InvivoGen, #ant-pr-1) for 3 days.

For generating cell spike-in standards, virus for five NT sgRNA plasmids (sgNT_5-9) possessing the 293T cell line barcode was produced in an arrayed format. 293T cells were transduced with the viruses individually. Transduced cells were selected using 2 μg/mL puromycin (InvivoGen, #ant-pr-1) for 3 days. To pool, cell lines were detached, resuspended in PBS, counted, and pooled at the following ratio: 1x sgNT_5, 3x sgNT_6, 9x sgNT_7, 27x sgNT_8, 81x sgNT_9. Pooled cells were aliquoted, pelleted at 425 × g for 5 min, and stored at −20 °C.

**Western blot analysis.** For assessing induction of FLAG-Cas9, MDA-MB-231 pLVX-TetOne Cas9 cells were treated with 0, 0.5, 1, 2, 5, or 10 ng/mL doxycycline (Clontech, #631311) for 48 h. Cells were harvested and lysed in buffer containing 50 mM Tris, pH 7.5, 1 mM EDTA, 150 mM NaCl, 1% Triton X-100, 0.1% SDS. Protein concentration was measured using the BCA Protein Assay Kit (Pierce, #23225). Five micrograms of total protein per lane was electrophoresed and transferred using an iBlot 2 Dry Blotting System (Thermo Fisher Scientific). Membrane was probed with 1:1000 Anti-FLAG primary antibody (Sigma-Aldrich, #F3165) followed by 1:10,000 Anti-Mouse IgG-Peroxidase secondary antibody (Sigma-Aldrich, #A9044), developed with Immobilon Western Chemiluminescent HRP Substrate (Millipore, #WBKLS0500), visualized using a ChemiDoc Touch Imaging System (Bio-Rad), and analyzed using Image Lab 5.2.1 (Bio-Rad). Membrane was stripped with ReBlot Plus Mild Antibody Stripping Solution (Millipore, #2502) and reprobed with 1:10,000 Anti-Alpha Tubulin primary antibody (Abcam, #ab80779) followed by 1:10,000 Anti-Mouse IgG-Peroxidase secondary antibody (Sigma-Aldrich, #A9044).

For confirming whole population knockout of the proteostasis factors, MDA-MB-231 pLVX-TetOne Cas9 cells transduced with appropriate sgRNAs were treated with 10 ng/mL doxycycline (Clontech, #631311) for 96 h (refreshing doxycycline every 2 days) to induce Cas9 expression prior to harvesting. Western blot analysis was performed as described above using the following antibodies: 1:1000 Anti-HSF1 (Santa Cruz Biotechnology, #sc-9144), 1:1000 Anti-HSF2 (Santa Cruz Biotechnology, #sc-13517), 1:1000 Anti-IRE1 (Cell Signaling Technology, #3294), 1:1000 Anti-XBP1 (Cell Signaling Technology, #12782), 1:1000 Anti-ATF3 (Abcam, #ab207434), 1:1000 Anti-ATF4 (Cell Signaling Technology, #11815), 1:1000 Anti-ATF6 (Cell Signaling Technology, #65880), 1:1000 Anti-ATG7 (Cell Signaling Technology, #8558), 1:1000 Anti-NRF2 (Cell Signaling Technology, #12721), 1:1000 Anti-KEAP1 (Cell Signaling Technology, #4617), 1:10,000 Anti-Alpha Tubulin (Abcam, #ab80779), 1:10,000 Anti-Beta Actin (Thermo Fisher Scientific, #MA5-15739). All uncropped blots are provided as a Source data file.

**Relative cell abundance competition experiment.** ZR-75-1, SKBR3, HCC-38, MDA-MB-231, and BT-20 cells were transduced with pHIV-Luc-ZsGreen (Addgene, Plasmid #39196) or pUltra-Chili-Luc (Addgene, Plasmid #48688) and sorted for GFP+ or RFP+ cells, as appropriate.

For preparing the five original pools, fluorescently labeled cell lines were counted, pooled, and frozen in liquid nitrogen. Pools were thawed on Day 0 and cultured normally. Six days and 13 days after thawing, the percentages of GFP+ and RFP+ cells were quantified by flow cytometry analysis using a LSRFortessa Cell Analyzer (BD Biosciences).

For estimating the growth rate (r) of each of the five cell lines, the equation for exponential cell growth was used:

$$x_t = x_0(1 + r)^t,$$

where $x_t$ = percentage at day t, $x_0$ = percentage at day 0, r = growth rate, and t = time [days]. Growth rates (r) were then used to calculate the optimal starting percentage ($x_0$) of each cell line to achieve 20% representation at t = 7 days.

For preparing the five optimized pools, fluorescently labeled cell lines were counted, pooled according to our model, and frozen in liquid nitrogen. Pools were thawed on Day 0 and cultured normally. Seven days after thawing, the percentages of GFP+ and RFP+ cells were quantified by flow cytometry analysis using a LSRFortessa Cell Analyzer (BD Biosciences). Heat maps were generated using Cluster 3.0 and Java TreeView 1.1.6r4.

**Cell pooling.** For preparing cell pools for QMAP-Seq with multiple cell lines (five cell lines × 12 genetic perturbations), individual cell line pools consisting of 12 sgRNAs (11 gene-targeting sgRNAs + 5 pooled NT sgRNAs) were prepared as

described in "Cell engineering". Each of the five individual cell line pools was counted, pooled with the other four cell line pools according to ratios derived from the relative cell abundance competition experiment, and frozen in liquid nitrogen (2,000,000 total cells per vial).

**Cell viability assays.** For confirming the efficacy of the sgRNAs that target SLC35F2, MDA-MB-231 pLVX-TetOne Cas9 cells engineered with either sgNT or one of the four sgRNAs targeting SLC35F2 were induced with 100 ng/mL dox-ycycline for 6 days (refreshing doxycycline every three days). In total, 1000 cells were seeded in a volume of 50 μL in opaque 384-well plates. The next day, YM155 was added over a nine-point concentration range in quadruplicate using a D300e Digital Dispenser (Tecan). Seventy-two hours later, cell viability was measured using the CellTiter-Glo Luminescent Cell Viability Assay (Promega, #G7572). Luminescence was read using an Infinite M1000 PRO (Tecan) with an integration time of 500 ms.

For live-cell imaging, pooled MDA-MB-231 pLVX-TetOne Cas9 cells engineered with 12 sgRNAs were induced with 100 ng/mL doxycycline for six days (refreshing doxycycline every 3 days). In total, 5000 pooled cells were seeded in a volume of 100 μL in 96-well plates. The next day, 89 compounds were added over the indicated four-point concentration range in duplicate from custom compound plates prepared at the High Throughput Analysis Laboratory (Northwestern University). Seventy-two hours later, the percent confluence of the population of cells in each well was measured (4X objective, whole-well imaging, phase channel) using an IncuCyte ZOOM Live-Cell Analysis System GUI version 2015A (Essen BioScience). To calculate relative cell confluence, the percent confluence of a compound-treated well was normalized to the median percent confluence of DMSO-treated wells.

For validating QMAP-Seq hits using Resazurin, MDA-MB-231 pLVX-TetOne Cas9 cells engineered with appropriate sgRNAs were induced with 100 ng/mL doxycycline for 6 days (refreshing doxycycline every 3 days). In total, 5000 cells were seeded in a volume of 100 μL in 96-well plates. The next day, Carfilzomib was added over the indicated four-point concentration range in triplicate using a D300e Digital Dispenser (Tecan). Seventy-two hours later, cell viability was measured using the Resazurin Cell Viability Kit (Cell Signaling Technology, #11884). Relative fluorescent units were read (excitation = 550 nm, emission = 605 nm) using an Infinite M1000 PRO (Tecan).

For all cell viability assays, dose–response curves were fit in GraphPad Prism 8 using the log(inhibitor) vs. response model (three parameters) with the top constrained to 100%.

**Compounds.** YM155 was obtained from Selleckchem (#S1130). Compounds for QMAP-Seq were obtained from the FDA-Approved Drug Library (MedChemExpress, #HY-L022), the Clinical Compound Library (MedChemExpress, #HY-L026), or from the following vendors: 4-Hydroxytamoxifen was obtained from Sigma-Aldrich (#H7904), and Bortezomib was obtained from Cayman Chemical (#10008822). For validation experiments, Carfilzomib was obtained from a different vendor (Cayman Chemical, #17554) than where it was sourced for QMAP-Seq.

**Selection of compounds and doses for QMAP-Seq.** Compounds for QMAP-Seq with multiple cell lines spanned six categories: positive controls, chemotherapeutics, targeted therapies (NCI), targeted therapies (OncoKB), diverse compounds (Informer Set), and proteostasis-modulating compounds. Relevant positive controls (4-Hydroxytamoxifen, Fulvestrant, Lapatinib, and YM155) were included. For all remaining compound categories, compounds that were not part of the FDA-Approved Drug Library (MedChemExpress, #HY-L022) or the Clinical Compound Library (MedChemExpress, #HY-L026) were filtered out. One compound per compound class was selected from breast cancer chemotherapeutics (Supplementary Fig. 3). Compounds that were classified as OncoKB Levels 1–4 for the indications of breast cancer, all solid tumors, or all tumors were selected from the OncoKB targeted therapies[60]. The existence of specific genetic alterations in breast cancer was confirmed using cBioPortal[61,62]. Up to two proteostasis-modulating compounds per target were incorporated.

Compounds were applied over a four-point concentration range (10-fold dilutions) in duplicate using one of two dose ranges. The standard dose range encompassed doses from 10 nM to 10 μM, and the low dose range covered doses from 1 nM to 1 μM. Dose ranges were selected by referencing doses previously used for treating cancer cell lines, including the Informer Set[22], Cancer Therapeutics Response Portal v2[21–23], Genomics of Drug Sensitivity in Cancer[18–20], and Selleckchem.

**QMAP-Seq**
*Induction, seeding, and compound treatment.* A step-by-step protocol describing the QMAP-Seq protocol can be found at Protocol Exchange[63]. For QMAP-Seq with multiple cell lines, pooled cell lines (five cell lines × 12 genetic perturbations) were thawed in media containing 100 ng/mL doxycycline (Clontech, #631311). Three days later, cells were expanded and doxycycline was refreshed. Three more days later, 5000 pooled cells were seeded in a volume of 100 μL in 96-well plates using an EL406 Microplate Washer Dispenser (BioTek Instruments), while

maintaining doxycycline induction. The next day, 180 compounds were added over a four-point concentration range in duplicate from custom compound plates prepared at the High Throughput Analysis Laboratory (Northwestern University). Briefly, compound plates containing 500 nL of compound at 1000× were resuspended in 250 μL media per well to achieve 2× compound concentration, and 100 μL of 2× compound was distributed to replicate wells using multichannel pipet. Cells were treated for 72 h.

*Lysis of cell spike-in standards.* Cell spike-in standards were thawed at room temperature for 5 min and resuspended in Lysis Buffer (10% 10x Taq DNA Polymerase Buffer (Invitrogen, #18067017), 0.45% IGEPAL CA-630 (Sigma-Aldrich, #I8896), 0.45% TWEEN 20 (Sigma-Aldrich, P9416), 10% Proteinase K (Qiagen, #19133), 79.1% Nuclease-Free Water (Qiagen, #129115)) to achieve a concentration of 36.3 total cell spike-in standards/μL. Cells were homogenized using a 5 mL syringe and a 21G × 1″ needle three times followed by a 27G × ½″ needle three times. Cells were incubated in 60 °C water bath for 1 h, pipetting up and down every 20 min.

*Lysis of compound-treated cells.* After 72 h of compound treatment, compound-treated cells were washed with 100 μL PBS. 50 μL Lysis Buffer containing 36.3 total cell spike-in standards/μL (1815 spike-in cells/well) was added using multichannel pipet. Plates were covered with foil adhesive and incubated in 60 °C oven for 1 h. Cell lysates were transferred to PCR plates using multichannel pipet. Proteinase K from Lysis Buffer was inactivated at 95 °C for 15 min using thermocycler.

*Sequencing library preparation.* A 359–367 bp fragment (depending on stagger length) containing the sgRNA and cell line barcode was amplified with a unique set of P5 and P7 primers for each well by combining the following reagents: 6 μL 10x Taq DNA Polymerase Buffer, 1.2 μL 10 mM dNTP Mix, 1.8 μL 50 mM MgCl₂, 0.6 μL DMSO (Sigma-Aldrich, #D8418), 6 μL 1 μM P5 Primer Mix, 10% Proteinase K, 6 μL 1 μM P5 Primer Mix (mix of 0-8nt staggered primers), 6 μL 1 μM P7 Primer, 8 μL Lysate (Proteinase K inactivated again just prior to PCR), 30.16 μL Water, 0.24 μL Platinum Taq DNA Polymerase (Invitrogen, #10966034). To reduce the likelihood of PCR jackpot effects, half of the PCR volume from each reaction was transferred to a second set of PCR plates. PCRs were run in thermocycler as follows: 94 °C for 4 min, (94 °C for 30 s, 60 °C for 30 s, 72 °C for 30 s, repeat for a total of 29 cycles), 72 °C for 15 min, 4 °C hold.

Technical duplicates were recombined. Library size was verified by running 5 μL of PCR products on E-Gel 96 2% Agarose gels (Invitrogen, #G7008-02). An equal volume (6 μL) of each PCR product was pooled together. The concentration of the pooled PCR products was measured using the Qubit dsDNA HS Assay Kit (Invitrogen, #Q32854). Pooled PCR products were purified using the QIAquick PCR Purification Kit (Qiagen, #28106) with sufficient PCR purification columns to avoid exceeding the maximum binding capacity of each column. The concentration of the purified sample was measured using the Qubit dsDNA HS Assay Kit (Invitrogen, #Q32854). The purified sample was purified a second time using the Purification Module with Magnetic Beads (Lexogen, #022.96). The final library concentration was measured using the Qubit dsDNA HS Assay Kit (Invitrogen, #Q32854).

*Next-generation sequencing.* The sequencing library was diluted to 2.5 nM, combined with PhiX (Illumina, #FC-110-3001) to achieve 25% PhiX (to increase nucleotide diversity), and denatured. The sequencing library was loaded on a NovaSeq 6000 (Illumina) using a NovaSeq 6000 S1 Reagent Kit, 200 cycles, 1.3B Reads (Illumina, #20012864). Single-end sequencing was performed using the following run parameters: Illumina Read 1 Primer: 164 cycles (to sequence sgRNA, cell line barcode), Illumina Index 1 Primer: 6 cycles (to sequence i7 index), Illumina Index 2 Primer: 6 cycles (to sequence i5 index).

*Sequencing data processing.* Individual samples were demultiplexed based on i5 and i7 index sequences (Supplementary Data 3 and 4) by running bcl2fastq2 Conversion Software v2.20 (Illumina). sgRNA and cell line barcode sequences were extracted from Read 1 sequences as follows: [ACACCG][sgRNA:20][Interval:102] [CellLineBarcode:8]. The number of reads for each cell line-sgRNA pair was counted, allowing up to one nucleotide mismatch per barcode. Next, cell number was interpolated from sequencing reads using sample-specific standard curves. The number of cells for a compound-treated sample was normalized to the median number of cells for the DMSO-treated samples.

Relative cell numbers were plotted as heat maps and dose–response curves. Heat maps were generated using Cluster 3.0 and Java TreeView 1.1.6r4. Dose–response curves were fit in GraphPad Prism 8 using the log(inhibitor) vs. response model (three parameters) with the top constrained to 100%. The area under the curve (AUC) was calculated using the equation: AUC = 0.5(dose 1 cell number) + dose 2 cell number + dose 3 cell number + 0.5(dose 4 cell number).

Data were filtered using the following exclusion criteria. First, samples with low read counts were excluded, which included these compounds: ABT199, Belinostat, BMS345541, Dexamethasone, Fingolimod, and RacRotigotine. Second, cell line-sgRNA pairs with high variation (standard deviation of log₂(cell number) > 0.7) were excluded, which included these cell line-sgRNA pairs: ZR-75-1 sgHSF1 and ZR-75-1 sgKEAP1. Third, compounds with an AUC > 550 for any cell line-sgRNA

pair were excluded, which included these compounds: Ruxolitinib and STF083010 (both in well B7, suggesting a likely technical problem with that well). Fourth, noncytotoxic cell line-compound pairs (relative cell number of sgNT with the highest dose ≥ 90%) were excluded, which included 225 cell line-compound pairs. Fifth, cell line-compound pairs without a significant dose-dependent reduction in cell viability (difference in relative cell number of sgNT between lowest and highest dose < 25%) were excluded, which included 366 cell line-compound pairs (union between fourth and fifth exclusion criteria = 380 cell line-compound pairs).

*QMAP-Seq with one cell line.* Experimental and analysis workflows were performed as described for QMAP-Seq with multiple cell lines with the following modifications. MDA-MB-231 cells with 12 genetic perturbations were used instead of five cell lines with 12 genetic perturbations each. Because each sample had five times as many cells per perturbation in the one cell line experiment (5000 cells ÷ 12 perturbations ≈ 417 cells per perturbation) compared to in the five cell line experiment (5000 cells ÷ 60 perturbations ≈ 83 cells per perturbation), five times as many cells for each of the cell spike-in standards were added per sample for the one cell line experiment. The data from Plate 8 was excluded from analysis due to a technical problem with the addition of cell spike-in standards for that plate. The sequencing library was loaded on a NextSeq 500 (Illumina) using a NextSeq 500/550 High Output Reagent Kit, 400M Reads (Illumina, #20024906).

**Network analysis**. For constructing a chemical–genetic interaction network, data were filtered as described above. All interactions were then filtered for those considered significant ($P < 0.05$) and that had a large effect size (absolute magnitude of AUC change >60). These interactions were visualized as an unweighted network using a standard force-directed layout in Cytoscape v3.7.2 (cytoscape.org).

For assessing functional similarity between proteostasis genes targeted in our sgRNA library, data were filtered as described above. The AUC difference was quantified for each genetic perturbation across all 488 cell line-compound contexts. A Spearman correlation was then calculated for all gene pairs based on overall similarity of their chemical–genetic interaction profiles.

**Statistical analysis**. Statistical analysis was performed with GraphPad Prism 8 statistical software. Replicate measurements were taken from distinct biological samples.

For correlation analysis of replicates, Pearson $r$ was reported, and statistical significance of Pearson correlation was determined using a two-tailed test ($n = 288$ compound-dose combinations after excluding Plate 8 data). For correlation analysis of AUCs between live-cell imaging and QMAP-Seq, Pearson $r$ was reported, and statistical significance of Pearson correlation was determined using a two-tailed test ($n = 89$ compounds). For correlation analysis of AUCs between two independent QMAP-Seq experiments, Pearson $r$ was reported, and statistical significance of Pearson correlation was determined using a two-tailed test ($n = 220$ compound-sgRNA combinations). For comparing the variation between plates for reads versus cell number, statistical significance was determined using an unpaired, two-tailed F test to compare variances ($n = 16$ plates).

For identifying the chemical–genetic interactions with the greatest effect, a volcano plot was produced after data filtering as described above. The magnitude was determined by calculating the difference in mean AUC between sgRNA and sgNT for every cell line-compound combination. The statistical significance was determined using an unpaired, two-tailed $t$ test ($n = 2$ biologically independent replicates). The chemical–genetic interactions with an Absolute AUC Difference > 25 and $P < 0.05$ were designated as hits.

For determining pathway enrichment, statistical significance was determined using a one-tailed binomial test to compare observed distribution to expected distribution ($n = 180$ compounds).

**Reporting summary**. Further information on research design is available in the Nature Research Reporting Summary linked to this article.

## Data availability

The data discussed in this publication, including raw fastq files, read counts, and relative cell numbers, have been deposited in NCBI's Gene Expression Omnibus and are accessible through GEO Series accession number GSE155855. Any other relevant data are available from the authors upon reasonable request. Source data are provided with this paper.

## Code availability

Custom code is available at GitHub (https://github.com/mendillolab/QMAP-Seq)[64] and Code Ocean (https://codeocean.com/capsule/3022355/tree/v1)[65].

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

## Acknowledgements
We thank the members of the Mendillo Lab, R. Smith, J. Choi, H. Liu, and J. Yu for suggestions and feedback. We thank L. Zou for helpful discussion. We thank S. Marshall, E. Rendleman, E. Clark, and D. Zha for sequencing support. Schematics were created with BioRender.com. This research was supported by grants from the Susan G. Komen Foundation CCR17488145, the National Cancer Institute of the NIH R00CA175293, and the Lynn Sage Cancer Research Foundation (to M.L.M). M.L.M was also supported by Kimmel Scholar (SKF-16-135) and Lynn Sage Scholar awards. D.R.A. was supported by 5T32GM008152-33. A part of this work was performed by the Northwestern University High Throughput Analysis Laboratory (NU-HTA), which is funded by the Cancer Center Support Grant P30 CA060553 from the National Cancer Institute awarded to the Robert H. Lurie Comprehensive Cancer Center. E.T.B. was supported by 5R50CA221848.

## Author contributions
S.B., G.W., and M.L.M. conceptualized the project and designed the assay. S.B. and G.W. engineered cell lines. M.R.C. arrayed compounds. S.B. and G.W. performed QMAP-Seq with one cell line. S.R.T. performed mathematical modeling. S.B. performed QMAP-Seq with multiple cell lines. S.B., G.W., and M.L.M. analyzed the data. E.T.B., J.M.J., and S.B. designed the bioinformatic analysis pipeline. E.T.B. and J.M.J. wrote the code for the bioinformatic analysis pipeline. S.B. performed validation experiments. D.R.A. performed network analysis. S.B. and M.L.M. wrote the manuscript. M.L.M. supervised the project.

## Competing interests
The authors declare no competing interests.
