## [Peer Review File · Nature Communications]

Reviewers' Comments:

Reviewer #1:

Remarks to the Author:

Mendillo and co-workers report the development, optimization, validation, and proof-of-principle application of a new method for assessing chemical genetic interactions. They term the method QMAP-Seq. Essentially, this is a next-gen sequencing approach to evaluate, in relatively high-throughput and at moderate expense, how cell viability phenotypes induced by large numbers of small molecules interplay with different genetic backgrounds created using Cas9 knockouts in multiple cell types. The authors apply their new method to look at how various compound treatments interplay with disruption of key proteostasis factors across five cancer cell lines.

I have no major criticisms of the methodology or the experiments presented. The work is very well-done, logically presented, and carefully interpreted. The significance of the work lies in the method development itself, not in any biological / mechanistic discovery, as the culminating experiment really does just provide a proof-of-principle dataset with no major finding associated. However, I think the significance and usefulness of the method itself, as well as the overall quality of the work, clearly justify publication in Nat Comm without any further biological follow-up. I expect to see other labs adopt the method upon publication.

A minor note: the discussion section would benefit from a more comprehensive evaluation of QMAP-Seq's limitations (e.g., difficult to use for assessing non-cell viability linked phenotypes) and areas for further development (perturbations that could be applied other than small molecule treatment, building searchable databases for QMAP-Seq data collection that are useful to the broader scientific community as data is generated on a large scale over time).

Reviewer #2:

Remarks to the Author:

This is an interesting manuscript that describes QMAP-Seq, a next generation sequencing-based method for studying chemical genetic interactions across multiple cell lines of multiple genotypes, treated with multiple chemical compounds. The work describes the method, its validation, and the results of screens that primarily serve to build confidence in the method, given that they recapitulate known compound-target and compound-resistance mechanistic interactions.

QMAP-Seq seems to be similar to the PRISM method, which also involves the pooling of multiple barcoded cell lines, with barcode abundance read out not by sequencing, but by Luminex bead hybridization. QMAP-Seq is attractive in that it uses readily available next generation sequencing, rather than Luminex technology, which is less ubiquitous in typical research labs. It's not clear, however, that the present work represents a conceptual advance as much as it's a technical improvement of an existing method that readers may indeed find useful. If there is a conceptual advance compared to PRISM, this should be pulled out more clearly in the text.

One important aspect of the current manuscript is the utilization of genetically engineered, isogenic cell lines as opposed to simply cell lines derived from different tumors. However, the use of isogenic lines is not tied to the QMAP-Seq method. That is, one could use pooled isogenic cell lines and read out drug sensitivity in other ways, or one could use QMAP-Seq with pools of non-isogenic lines. The manuscript would benefit from clarifying this.

The biological findings described in the screen are largely of a validation nature, rather than being novel in their own right. However, for this method-focused paper, this seems appropriate, and the work shouldn't be penalized for lack of novelty of biological findings.

In sum, the work appears to have been carefully conducted, and it represents a potentially useful

technical advance -- in particular because it is a method that in principle could be practiced by multiple labs without highly specialized instrumentation.

Point-by-Point Response

Reviewer #1

Mendillo and co-workers report the development, optimization, validation, and proof-of-principle application of a new method for assessing chemical genetic interactions. They term the method QMAP-Seq. Essentially, this is a next-gen sequencing approach to evaluate, in relatively high-throughput and at moderate expense, how cell viability phenotypes induced by large numbers of small molecules interplay with different genetic backgrounds created using Cas9 knockouts in multiple cell types. The authors apply their new method to look at how various compound treatments interplay with disruption of key proteostasis factors across five cancer cell lines.

I have no major criticisms of the methodology or the experiments presented. The work is very well-done, logically presented, and carefully interpreted. The significance of the work lies in the method development itself, not in any biological / mechanistic discovery, as the culminating experiment really does just provide a proof-of-principle dataset with no major finding associated. However, I think the significance and usefulness of the method itself, as well as the overall quality of the work, clearly justify publication in Nat Comm without any further biological follow-up. I expect to see other labs adopt the method upon publication.

A minor note: the discussion section would benefit from a more comprehensive evaluation of QMAP-Seq's limitations (e.g., difficult to use for assessing non-cell viability linked phenotypes) and areas for further development (perturbations that could be applied other than small molecule treatment, building searchable databases for QMAP-Seq data collection that are useful to the broader scientific community as data is generated on a large scale over time).

Author Response: Thank you for these excellent suggestions. We agree that assessing non-cell viability linked phenotypes is a potential limitation of QMAP-Seq. We have added this to the discussion of QMAP-Seq's limitations together with the possibility of isolating the desired phenotype prior to sequencing as a way to overcome this limitation on lines 350-352 of the revised manuscript. We have also incorporated a discussion of additional perturbations that could be applied on lines 362-364 and the exciting possibility of building searchable QMAP-Seq databases on lines 364-366.

Reviewer #2

This is an interesting manuscript that describes QMAP-Seq, a next generation sequencing-based method for studying chemical genetic interactions across multiple cell lines of multiple genotypes, treated with multiple chemical compounds. The work describes the method, its validation, and the results of screens that primarily serve to build confidence in the method, given that they recapitulate known compound-target and compound-resistance mechanistic interactions.

QMAP-Seq seems to be similar to the PRISM method, which also involves the pooling of multiple barcoded cell lines, with barcode abundance read out not by sequencing, but by Luminex bead hybridization. QMAP-Seq is attractive in that it uses readily available next generation sequencing, rather than Luminex technology, which is less ubiquitous in typical research labs. It's not clear, however, that the present work represents a conceptual advance as much as it's a technical improvement of an existing method that readers may indeed find useful. If there is a conceptual advance compared to PRISM, this should be pulled out more clearly in the text.

Author Response: Thank you for pointing out the need to differentiate our method from the PRISM method in the text. We have incorporated these valuable suggestions on lines 320-327 of the revised manuscript. In brief, we now point out that unlike PRISM, which only multiplexes cell lines, QMAP-Seq also multiplexes thousands of compound treatments (whereas each compound treatment in PRISM is an individual assay). QMAP-Seq also leverages ubiquitous (and continually improving) next-generation sequencing technology. As you mention below, another distinguishing feature of QMAP-Seq is the use of

isogenic cell line pairs, which we agree is an experimental design decision that could also be performed using PRISM.

One important aspect of the current manuscript is the utilization of genetically engineered, isogenic cell lines as opposed to simply cell lines derived from different tumors. However, the use of isogenic lines is not tied to the QMAP-Seq method. That is, one could use pooled isogenic cell lines and read out drug sensitivity in other ways, or one could use QMAP-Seq with pools of non-isogenic lines. The manuscript would benefit from clarifying this.

Author Response: These are great points. We have added language on lines 322-325 of the revised manuscript to clarify that although using isogenic cell lines is an attribute of QMAP-Seq, QMAP-Seq does not require the use of isogenic cell lines nor is it the only possible method for performing pooled drug sensitivity assays using isogenic cell lines.

The biological findings described in the screen are largely of a validation nature, rather than being novel in their own right. However, for this method-focused paper, this seems appropriate, and the work shouldn't be penalized for lack of novelty of biological findings.

In sum, the work appears to have been carefully conducted, and it represents a potentially useful technical advance -- in particular because it is a method that in principle could be practiced by multiple labs without highly specialized instrumentation.